Improved MobileNetV2 crop disease identification model for intelligent agriculture

Lu Jianbo 1 2
Liu Xiaobin 1
Ma Xiaoya 2 3 maxy519@126.com
Tong Jin 3
Peng Jungui 1
1 School of Computer and Information Engineering, Nanning Normal University , Nanning, Guangxi , China
2 Guangxi Key Lab of Human-machine Interaction and Intelligent Decision, Nanning Normal University , Nanning, Guangxi , China
3 School of Logistics Management and Engineering, Nanning Normal University , Nanning, Guangxi , China
Murugappan M.
Electronic publication date: 2023 Sep 25
Publication date: 2023
Volume: 9
Electronic Location ID: e1595
Received 2023 May 15; Accepted 2023 Aug 27
Copyright: © 2023 Lu et al.
Copyright year: 2023
Copyright holder: Lu et al.
License: This is an open access article distributed under the terms of the Creative Commons Attribution License, which permits unrestricted use, distribution, reproduction and adaptation in any medium and for any purpose provided that it is properly attributed. For attribution, the original author(s), title, publication source (PeerJ Computer Science) and either DOI or URL of the article must be cited.
License URL: https://creativecommons.org/licenses/by/4.0/

Keywords: Intelligent agriculture, Crop disease identification, Lightweight, MobileNetV2, RepMLP

Funding: Guangxi Key R&D Project AB21076021 Project of Humanities and Social Sciences of “Cultivation Plan for Thousands of Young and Middle-aged Backbone Teachers in Guangxi Colleges and Universities” 2021 Research on Collaborative Integration of Logistics Service Supply Chain Under High-quality Development Goals 2021QGRW044 This study was supported by the Guangxi Key R&D Project (Grant Numbers Gui Ke AB21076021), the Project of Humanities and Social Sciences of “cultivation plan for thousands of young and middle-aged backbone teachers in Guangxi Colleges and universities” in 2021: and the Research on Collaborative integration of logistics service supply chain under high-quality development goals (Grant Numbers 2021QGRW044). The funders had no role in study design, data collection and analysis, decision to publish, or preparation of the manuscript.

==============================
Using intelligent agriculture is an important way for the industry to achieve high-quality development. To improve the accuracy of the identification of crop diseases under conditions of limited computing resources, such as in mobile and edge computing, we propose an improved lightweight MobileNetV2 crop disease identification model. In this study, MobileNetV2 is used as the backbone network for the application of an improved Bottleneck structure. First, the number of operation channels is reduced using point-by-point convolution, the number of parameters of the model is reduced, and the re-parameterized multilayer perceptron (RepMLP) module is introduced; the latter can capture long-distance dependencies between features and obtain local a priori information to enhance the global perception of the model. Second, the efficient channel-attention mechanism is added to adjust the image-feature channel weights so as to improve the recognition accuracy of the model, and the Hardswish activation function is introduced instead of the ReLU6 activation function to further improve performance. The final experimental results show that the improved MobilNetV2 model achieves 99.53% accuracy in the PlantVillage crop disease dataset, which is 0.3% higher than the original model, and the number of covariates is only 0.9M, which is 59% less than the original model. Also, the inference speed is improved by 8.5% over the original model. The crop disease identification method proposed in this article provides a reference for deployment and application on edge and mobile devices.

Introduction

The issue of agricultural production is of great significance, and it has been widely studied by scholars in many fields around the world. Intelligent agriculture takes information, knowledge, and equipment as its core elements. Through deep cross-border integration of information technology, intelligent equipment, and agriculture, it has promoted the improvement of the comprehensive management and control of agricultural systems. The research around agricultural development has led to continuous improvement of intelligent agriculture (Su & Wang, 2021; Hou et al., 2021; Liu et al., 2021). In the field of agricultural production, identification of crop diseases and pests is an important means for crop management and control, and it can help to ensure the yield and quality of agricultural products. In intelligent agriculture, environmentally friendly and efficient techniques for identification and prevention of crop diseases are important for ensuring crop production and the quality of agricultural products. With continuous increases in the needs and requirements of food production, developments in science and technology have led to ongoing improvements in methods for identifying crop diseases.

In recent years, the rapid development of artificial intelligence has opened new avenues of research in identification of crop diseases (Zhai et al., 2021). Li, Ma & Wang (2012) proposed a support vector machine (SVM) approach using multiple feature parameters for the identification of two diseases of wheat. They first cropped sub-images from typical images of disease features, then used a median-filtering algorithm for image denoising, applied a k-means clustering algorithm for extraction of disease features, and finally used an SVM algorithm for classification tests; in their experimental results, the achieved a 100% test-sample recognition rate. Wei et al. (2017) used an SVM-based method for multiclassification detection of four plant foliar diseases. They used a lab color-space model to segment and extract leaf-disease features, then used k-means clustering to enhance the segmentation clustering effect. Finally, they applied the SVM algorithm for classification. The recognition rate of their method reached 89.5%.

Although there has been some progress in research around SVM models in this field, some of the methods that are applied can require additional algorithms to extract the features of plant leaf diseases before the SVM algorithm can be used for classification and identification. The process of this research into algorithms is cumbersome, and it falls short of meeting the requirements of practical applications when there is a need for classification of multiple plant leaf diseases. The emergence of convolutional neural networks (CNNs) has solved the problem of the tedious and inefficient process of feature extraction from crop disease images. Consequently, CNNs have become the main algorithmic construct in the field of image recognition. Li et al. (2020) proposed a model for eggplant disease identification. They used an SE-Inception structure with a multiscale feature-extraction module to extract disease features. The top-1 and top-5 accuracy values obtained using this system reached 99.27% and 99.99%, respectively, on a dataset of four eggplant leaf diseases, and the model had 14.8 M parameters. Du et al. (2012) proposed an improved ResNet model for identifying the degree of crop disease damage. Their improved ResNet model introduced an effective channel-attention mechanism and also incorporated a multiscale fusion strategy in the classification layer. These methods enhanced the feature-extraction ability of the model and improved its classification ability for fine-grained features; the final model, which had 53.9 M parameters, achieved an accuracy of 94.5%. Too et al. (2019) evaluated the performance of several common CNNs for plant disease classification. Their experiments on the PlantVillage public dataset showed that DenseNet achieved state-of-the-art performance with fewer parameters and computation time, with 99.75% accuracy and 7.1 M model parameters. To summarize, according to the results of recent research in this field, crop disease identification methods based on CNNs (Li et al., 2020; Du et al., 2012) have generally achieved remarkable results in terms of accuracy.

CNN-based models often have large numbers of parameters and are highly computationally intensive; they are therefore usually deployed using cloud-computing platforms. However, in some remote agricultural areas, due to weak network infrastructure, it can be difficult for devices to interact with cloud-based services to meet the demand for low-latency and low-cost crop disease identification. In such cases, it is appropriate to deploy neural network (NN) models for inference using mobile devices or edge-computing scenarios. However, mobile and edge devices, which are the main kind of devices used for supervision in agriculture, have limited computing and storage resources. This leads to difficulties in deploying parametric or computationally intensive models, and this in turn greatly limits the development of smart agriculture. Therefore, an efficient and lightweight network model is particularly important for development in the field of crop disease identification. Seeking to address the above problems, this research makes the following contributions.

(1) To achieve the highest yields and ensure the quality of crops, an improved MobileNetV2 model is proposed to expand the field of smart agricultural production by taking the limitations of edge and mobile devices as the entry point.

(2) Our proposal of this improved MobileNetV2 model seeks to solve the problem of accurate crop disease recognition in cases of limited computing resources. Compared with the original model, the improved MobileNetV2 model not only ensures recognition accuracy but also greatly reduces the number of parameters and computation required; it also improves the inference speed.

The remainder of this article is arranged as follows. The Related Work section presents a review of relevant literature. The Model architecture section gives details of the design of the model. The Experiments section describes our experimental verification of the model. Finally, the Conclusions provides a summary of our conclusions.

Related work

Intelligent agriculture

The development of a new generation of information technologies, such as big data, artificial intelligence, and the Internet of Things (IoT), has promoted the gradual transformation of traditional agriculture into intelligent agriculture (Liu et al., 2021). For example: Bagheri (2021) studied the application of remote sensing technology based on the IoT and big data in intelligent agriculture; Whig et al. (2022) studied the application of intelligent agriculture based on Fog IoT; Gao, Zhong & Liu (2021) conducted research into multi-source heterogeneous data-processing methods in intelligent agriculture using a cloud platform.

In intelligent agriculture, sensor technology can be used to collect various information relating to agricultural production, communication technology can transmit the collected information, and big-data analysis can be used to guide agricultural production and the marketing of agricultural products. The application of artificial intelligence can make agriculture highly productive and efficient, and driverless technology enables automated production and fine-tuned management. In particular, the application of key artificial-intelligence technologies in the field of agricultural production has become an area of intensive research. Because agricultural production is related to national economies and people’s livelihoods, it is a worldwide concern; production is also a fundamental problem in the field of agriculture, and smart agriculture is an inevitable trend in agricultural development. Hadidi, Saba & Sahli (2021) studied the role of artificial NNs in intelligent agriculture through a case study; Kashyap et al. (2021) studied a batch smart-irrigation system for precision agriculture using deep-learning NNs. The use of artificial NNs in deep learning is one of the key approaches in the field of smart agricultural production; they are often applied to the identification and detection of crop diseases to ensure the yield and quality of crop production through detection and prevention. For example, Chandy (2019) used deep learning to conduct research on pest identification in coconut trees, and Kong et al. (2022) studied the application of a spatial-feature-enhanced attention NN with high-order pooling representation for pest identification.

Identification of crop diseases

Crop disease is a problem that cannot be eliminated, and it is unavoidable in the production process of agricultural products. Crop disease image recognition is the comprehensive use of image processing, phytopathology, pattern recognition, and other technical means to analyze crop disease information. It seeks to obtain a feature representation and classification model of a disease to accurately identify it using images. Fast and accurate detection of plant diseases is essential for the sustainable improvement of agricultural productivity (Ngugi, Abelwahab & Abo-Zahhad, 2021). Traditionally, people have relied on experts to diagnose crop abnormalities caused by disease, pests, nutrient deficiencies, or extreme weather; however, relying on manual diagnosis is not only expensive and time-consuming, but there may also be limited expert resources available, and this can make it impractical in some cases. To cope with the challenges presented by crop anomalies, studying the identification of plant diseases using image-processing technology has become an intensive area of research.

Although the use of traditional image-processing methods for crop disease image recognition has achieved some results, these methods still have shortcomings; they can require complex preprocessing, they are often time-consuming and labor-intensive, and it is difficult to objectively and completely describe the image information required for disease identification. In recent years, research in the field of crop disease identification using image processing and machine-learning technology has progressed toward the use of deep NNs. Mique & Palaoag (2018) studied the detection of rice diseases and pests using CNNs; Lee, Lin & Chen (2020) used CNNs to identify leaf diseases and pests of tea leaves under actual field conditions; Agarwal et al. (2019) studied the detection of corn diseases based on a CNN.

The application of various electronic, information, and control technologies in machinery and equipment is crucial for the realization of the goal of smart agriculture. With the continuous development and application of smartphones, the management of crop pests and diseases (Che’Ya et al., 2022) has been supported by mobile application technologies and devices, and the demand for mobile applications in agriculture is increasing. Through mobile applications, farmers can find infections at an early stage and take specific treatment measures to prevent further infections. To make crop disease identification more convenient, accurate, and efficient, researchers have proposed lightweight NN models such as MobileNet (Howard et al., 2017; Sandler et al., 2018), Xception (Chollet, 2017), and ShuffleNet (Zhang et al., 2018; Ma et al., 2018). These systems greatly reduce the number of model parameters and computational intensity without significantly reducing recognition accuracy, providing support for their deployment on edge and mobile devices.

Lightweight networks have been verified in terms of both feasibility and mobile deployment for crop disease identification. Liu, Feng & Wang (2019) combined MobileNet and Inception v3, two lightweight CNNs for migration learning, and achieved 95.02% and 95.62% accuracy on the PlantVillage dataset, respectively; they showed that MobileNet is more suitable for mobile plant disease identification applications in terms of accuracy, model inference speed, and model size. Tang et al. (2020) used ShuffleNetV1 and ShuffleNetV2 as the backbone and introduced an attention mechanism; they achieved good recognition accuracy on a public grape leaf disease dataset. Taking MobileNetV2 as the backbone, Chen et al. (2021) introduced an attention mechanism to conduct pre-training on ImageNet; they then performed migration learning, which achieved good results on a public rice disease dataset.

The above studies show that the application of deep NNs to the identification of crop diseases and pests has achieved remarkable results, and CNNs stand out in this field. To reconcile the conflict between the limited computing resources of edge and mobile devices and the large amount of calculation required for these models, various lightweight networks for crop disease identification have been proposed and widely used. However, these models generally use or introduce an attention mechanism to improve and change the model structure; such approaches use only convolution for local feature extraction, and they lack the ability to extract global features. Based on this, we propose a lightweight crop disease identification method using an improved version of MobileNetV2.

Our model increases the global perception of the feature map by introducing the re-parameterized multilayer perceptron (RepMLP) module (Ding et al., 2021) to better capture long-range dependencies, which complements the local feature-extraction capability of the convolution module, while adding the efficient channel attention (ECA) mechanism to improve the accuracy of the model. The PlantVillage dataset includes images of 14 crops. Among these, leaves showing three or more states are included for five crops: apple, corn, grape, potato, and tomato, and 25 types of disease are included. Using this dataset, we tested the performance of the model for identifying a single crop presenting multiple disease characteristics. If crops have multiple diseases, detecting this and controlling them in time to improve yield and quality is an important task. Our lightweight model using the improved MobileNetV2 system was tested in experiments identifying the 25 disease categories in the five noted crops from the PlantVillage dataset.

Model architecture

This section describes the enhanced MobileNetV2 model. This decreases the number of model parameters by reducing the number of channels processed through the use of point-by-point convolution. The RepMLP and ECA modules are also introduced to enhance the global perception and expression capability of the model for features, which ultimately boosts its overall performance.

MobileNetV2 and ECA module

Depthwise separable convolution

Depthwise separable convolution includes deep convolution and point-by-point convolution, as shown in Fig. 1. Unlike standard convolution, in deep convolution, a single convolution kernel only computes one channel, and channel fusion is then performed using point-by-point convolution; this greatly reduces the computational effort required. Suppose the input characteristic image size is H × W, the convolution kernel size is K × K, the number of output characteristic channels is C, and the size of each output characteristic channel is N. The computation of the standard convolution is then:

Figure 1 (A) Standard convolution and (B) depthwise separable convolution.

(1) F=K×K×C×N×H×W

The computation of the depthwise separable convolution is:

(2) FDS=K×K×C×H×W+C×N×H×W

The ratio of the number of calculations in standard convolution to that in depthwise separable convolution is:

(3) FDSF=K×K×C×H×W+C×N×H×WK×K×C×N×H×W=1N+1K2.

According to the above formula, when the size of our convolution kernel is 3 × 3 and the number of output characteristic channels N is large, the number of calculations required for depthwise separable convolution is about 1/9th of that required for the standard convolution.

Inverted residuals

A schematic of the inverted residual module is shown in Fig. 2. Its structure serves the same purpose as a traditional residual structure: to solve the gradient-disappearance problem of the NN during backpropagation. However, the traditional residual structure seeks to reduce the dimension of the feature channel and then increase it again after feature extraction. In contrast, the inverse residual structure first increases the dimension of the feature channel and then reduces it again after feature extraction. The inverse residual module cancels the nonlinear activation function after the last 1 × 1 convolution. This is because the use of nonlinear activation from high to low dimensions will lead to the loss of information.

Figure 2 Structure diagram of the inverted residual module.

ECA net

Wang et al. (2020) proposed the ECA mechanism, which can achieve local cross-channel interaction efficiently via one-dimensional convolution. It can significantly enhance the performance of the model while adding only a few additional parameters and computations. Unlike the channel-attention mechanism of a squeeze-and-excitation network (Jie et al., 2017), the ECA mechanism uses a local cross-channel-interaction strategy without dimensionality reduction. This effectively avoids the effect of the dimensionality-reduction operation on the channel-attention learning effect. Furthermore, to avoid manually adjusting k by cross-validation, it includes a k-adaptive method whereby the value of k is proportional to the channel dimensionality.

The structure of the ECA module is shown in Fig. 3. First, feature compression is performed along the spatial dimension by a global average pooling (GAP) operation to transform the W × H × C feature map into a 1 × 1 × C vector. Then, one-dimensional convolution is used to complete the cross-channel information interaction, and the size of the convolution kernel is adaptively changed using a function so that a layer with a larger number of channels can interact more across channels. Finally, the normalized weights 1 × 1 × C are multiplied with the original input W × H × C feature map channel by channel to generate a weighted feature map, in which σ denotes the Sigmoid function.

Figure 3 ECA attention mechanism module.

RepMLP and parameter reconstruction

Ding et al. (2021) proposed a multilayer perceptron-style NN module for image recognition, called RepMLP, which consists of a series of fully connected (FC) layers. It leverages the global modeling and position-awareness features of FC layers, as well as the local-structure-extraction ability of convolution, to efficiently model long-distance and position patterns. The RepMLP module can enhance the global perception ability of the model and improve its overall performance.

The RepMLP module is composed of three parts: global perceptron, partition perceptron, and local perceptron. The global perceptron splits the image, and then each split image passes through the average pooling layer and two FC layers; the reconstructed dimension is then added to the split image. The partition perceptron reduces the dimension to decrease the number of parameters, and it then rebuilds the original dimension after passing through the FC layer. The local perceptron convolves the split image through multiple convolution layers, and it is then added to the output of the partition perceptron after batch normalization (BN). The structure of the RepMLP module is shown in Fig. 4, in which N is the batch size, C is the number of input channels, H and W are the height and width of the output, h and w are the characteristic height and width after splitting, g is the number of groups, p is the number of fills, and o is the number of output channels.

Figure 4 RepMLP module.

When the RepMLP module is deployed, convolution can be integrated into the FC layer, which speeds up the model inference; the reasoning behind this was discussed in a previous report (Ding et al., 2021). After parameter reconstruction in the RepMLP module, as shown in Fig. 5, the convolution (conv) and BN layers in the local perceptron are converted into three equivalent FC layers.

Figure 5 After refactoring the RepMLP module parameters.

Modified MobileNetV2

MobileNetV2 has seven Bottleneck structures, and each of these consists of three main parts: the first is a 1 × 1 point-by-point convolution for the up-dimensional operation; the second is a 3 × 3 depthwise separable convolution for feature extraction in higher dimensions, thus extracting richer feature information; the third is 1 × 1 convolution to drop back to the original dimension. Each of these parts has convolution operations between them and also contains BN and activation-function layers; however, the third part only has BN layers and does not contain activation-function layers. After the seven Bottleneck layers mentioned above, it is then up-dimensioned by a standard convolution, and this is followed by average pooling. Finally, the output results are passed through a linear layer as a classifier.

The majority of MobileNetV2’s model parameters are point-by-point convolutional ascending and descending operations. The number of model parameters can be reduced by decreasing the number of 1 × 1 convolution operations or by reducing the number of channels of these operations. When the number of channels is reduced, the features cannot be extracted in the high-dimensional space, and the 3 × 3 depthwise separable convolution operation must be used to obtain richer information about feature dimensions. It is then necessary to find new modules for feature extraction at lower latitudes or in a dimensional space with a smaller number of channels than the original. We therefore need to improve the Bottleneck structure.

The first part performs the channel-number-reduction operation using point-by-point convolution, and the number of channels is reduced to r times the original number (where r = 2, 4, 8). The second part uses the RepMLP module for feature extraction; the third part then raises the number of channels back to its original dimension using point-by-point convolution. The improved Bottleneck structure reduces the number of channels for the 1 × 1 convolution operation, resulting in a significant reduction in the overall number of network parameters. We named the Bottleneck module with RepMLP “Bottleneck_RepMLP”. The improved MobileNetV2 model is named MobileNet-RepMLP. The structure of MobileNet-RepMLP is shown in Fig. 6A.

Figure 6 MobileNet-RepMLP model structure.

The RepMLP module can obtain feature information in low-latitude space, and it compensates the local feature-extraction capability of the FC layer mainly through the combination of the FC layer and a convolutional layer. It also enables the FC layer to acquire local a priori information and capture long-range dependencies, thus enhancing the global-sensing capability of the model and enabling the acquisition of richer feature information. To further enhance the expressiveness of the model and improve its recognition accuracy, the ECA module is introduced after each Bottleneck and Bottleneck_RepMLP structure. The Hardswish activation function is used instead of the ReLU6 activation function in the Bottleneck_RepMLP structure. The structure of MobileNet-RepMLP after adding the ECA module and the Hardswish activation function is shown in Fig. 6B. The introduction of the ECA module can cause it to focus more on disease characteristics, reduce the interference of background and other factors, and improve its recognition accuracy; the increase in parameters and computational load caused by the inclusion of the ECA module is very small and indeed almost negligible. The Hardswish activation function provides smoothing and non-monotonic properties, and it outperforms the ReLU6 activation function on deep models; it therefore improves the overall performance of the model.

The structure of the MobileNet-RepMLP model is shown in Table 1, in which t denotes the number of feature channels expanded or reduced by a multiple, c denotes the number of feature channels, and s denotes the step size. The performance of Bottleneck_RepMLP on the CIFAR-10 dataset at different stages of introduction will be discussed in “PlantVillage experiment”.

Table 1 MobileNet-RepMLP network structure.

Input	Operator	t	c	n	s	
3 × 224 × 224	Conv2d 3 × 3	–	32	1	2	
32 × 112 × 112	Bottleneck1	1	16	1	1	
16 × 112 × 112	Bottleneck2	6	24	2	2	
24 × 56 × 56	Bottleneck3	6	32	3	2	
32 × 28 × 28	Bottleneck4	6	64	4	2	
64 × 14 × 14	Bottleneck_RepMLP1	1/4	96	3	1	
96 × 14 × 14	Bottleneck_RepMLP2	1/4	160	3	1	
160 × 7 × 7	Bottleneck_RepMLP3	1/4	320	1	1	
320 × 7 × 7	Conv2d 1 × 1	–	1,280	1	1	
1,280 × 7 × 7	Avgpool 7 × 7	–	–	1	–	

Experiments

The experimental environment configured for the experiment using Anaconda is shown in Table 2 shown. Each image in the CIFAR-10 dataset was padded to 40 × 40 size by data augmentation. The batch size was set to 128 and the epoch was set to 200. The learning rate was scheduled by cosine annealing, with an initial value of 0.1 and a final value of 0.0001 after 200 epochs. Each image in the PlantVillage dataset had a size of 224 × 224. The optimizer was SGD, the batch size was 64 and the epoch was 30. The learning rate was scheduled by cosine annealing, with an initial value of 0.1 and a final value of 0.0001 after 30 epochs.

Table 2 Detailed information of experimental environment.

Environment	Configuration parameters	
GPU	NVIDIA Quadro P2200 (5 GB/Intel)	
Framework	Pytorch3.8	
Python	Python3.7	
GPU acceleration library	CUDA10.2 CUDNN10.2	

Datasets and pre-processing

The CIFAR-10 data contains 60,000 color images from 10 categories in total, with 6,000 images per category. We used 50,000 images for training and 10,000 images for testing. The PlantVillage dataset (Hughes & Salathe, 2015) is a database of plant disease images that is open to all users. We selected crops whose leaves showed three or more states; this included five crops and 25 types of diseased leaves. Some representative images from the PlantVillage dataset are shown in Fig. 7.

Figure 7 PlantVillage partial image data.

As can be seen from the details in Table 3, the initial data distribution was extremely unbalanced; it therefore needed to be expanded. Data expansion, a data-enhancement technique, was performed on the divided dataset. A random combination of data-enhancement techniques such as flipping, mirroring, randomly adjusting image color (contrast, brightness), and adding noise were separately applied to the initial data. As shown in Fig. 8, the original Apple scab image can be flipped to get the enhanced image Apple scab_a, and it can also be rotated, have noise added, and its exposure can be adjusted to get the enhanced image Apple scab_b. The other leaves were also adjusted by similar random combinations of operations. After data expansion using these data-enhancement techniques, the training set reached about 1,500 sheets per class, totaling 37,572 sheets, and the test set reached 400 sheets per class, totaling 10,359 sheets. The training and test sets were divided in the ratio 4:1, and the distribution of these datasets is shown in Table 3.

Table 3 Image information of PlantVillage dataset.

Data category	Original data/sheet	Training set/sheet	Test set/sheet	
Apple scab	630	1,512	437	
Apple black rot	621	1,491	418	
Apple rust	275	1,510	416	
Apple healthy	1,645	1,495	404	
Corn gray spot	513	1,505	408	
Corn rust	1,192	1,502	409	
Corn healthy	1,162	1,501	402	
Corn leaf blight	985	1,506	394	
Grape black rot	1,180	1,500	436	
Grape black measles	1,383	1,507	424	
Grape healthy	423	1,508	402	
Grape leaf blight	1,076	1,506	415	
Potato early blight	1,000	1,500	400	
Potato healthy	152	1,464	420	
Potato late blight	1,000	1,500	400	
Tomato health	1,591	1,503	406	
Tomato spot blight	1,000	1,517	401	
Tomato two spotted spider mite	1,676	1,506	405	
Tomato late blight	1,909	1,528	401	
Tomato leaf mold	952	1,502	380	
Tomato bacterial spot	2,127	1,500	425	
Tomato target spot	1,404	1,514	487	
Tomato early blight	1,000	1,500	425	
Tomato mosaic virus	373	1,495	444	
Tomato yellow leaf	5,375	1,500	400	
Total	30,644	37,572	10,359	

Figure 8 Image enhancement operation.

RepMLP phase experiment

The CIFAR-10 dataset was used for these experiments. We used Bottleneck_RepMLP in different phases of MobileNetV2. MobileNetV2 has a total of seven Bottlenecks, and in the experiments, Bottleneck_RepMLP was progressively applied to Bottlenecks 4 to 7. The number of feature channels was uniformly reduced by a factor of 4 (r = 4), and the local perceptron convolution was carried out with K = 1, 3, and 5. The different experiments are shown in Table 4, in which Bottlenecks 4, 5, 6, and 7 are abbreviated as B4, B5, B6, and B7. The experiments listed in Table 4 show that by using Bottleneck_RepMLP, not only can the number of parameters be greatly reduced, but the accuracy can also be maintained; furthermore, higher accuracy than that obtained from the original model can be achieved on the CIFAR-10 dataset. The accuracy of the Mobilenet-RepMLP1 model was higher than that of the original model, and the number of parameters was only 0.55 M. After this, as the number of Bottleneck_RepMLP modules was reduced in order, and the accuracy of each of these models was slightly better than that of MobilenetV2, except for Mobilenet-repMLP4, which was less accurate than the original model. Among the models, the most accurate performance was found with Mobilenet-RepMLP2.

Table 4 The experimental results of improved Bottleneck usage stage.

Models	B4	B5	B6	B7	Accuracy/%	Param/M	
MobileNet-RepMLP1	√	√	√	√	86.89	0.55	
MobileNet-RepMLP2		√	√	√	86.75	0.73	
MobileNet-RepMLP3			√	√	86.40	1.0	
MobileNet-RepMLP4				√	85.02	1.7	
MobileNetV2					86.18	2.2	

In the next experiment, we compared the performance using the CIFAR-10 dataset between models using different scaling channel multipliers r. Referring to the experimental data in Table 4, Bottleneck_RepMLP was used uniformly in stages B5, B6, and B7. The experimental results are shown in Table 5. They show that on the CIFAR-10 dataset, varying the value of r results in relatively little difference in the number of parameters, but there is a difference in accuracy. To better make a better trade-off between the model performance and accuracy, the MobileNet-RepMLP model uses the Bottleneck_RepMLP structure in phases B5, B6, and B7 by default, while the r value was taken as 4.

Table 5 Experimental results of channel reduction factor.

Models	r	Accuracy/%	Param/M	
MobileNet-RepMLP2	8	86.66	0.69	
MobileNet-RepMLP2	4	86.75	0.73	
MobileNet-RepMLP2	2	86.56	0.81	

PlantVillage experiment

To better evaluate the performance of this improved model, the PlantVillage dataset was also tested with other models that are widely applied and considered excellent today, and the results were compared. The tested models were: ResNet18 (He et al., 2016), ShuffleNetV2 (Liu, Feng & Wang, 2019), and MobileNetV3 (Howard et al., 2019). It also includes the optimized ShuffleNet-ECANet 2.0 model (Li, Qiu & Zhang, 2022). The model is optimized by Li, Qiu & Zhang (2022) based on the ShuffleNet V2 model and has achieved good results on the PlantVillage crop disease dataset. The results of this experimental comparison are shown in Figs. 9, 10 and Table 6. They show that the accuracy and recall of our MobileNet-RepMLP model are 99.0% and 99.01%, respectively; these values are slightly lower than those from the other models, but the number of parameters of our model is only 0.91M. Compared with the lightweight networks MobileNetV2, MobileNetV3, and ShuffleNetV2, the number of parameters is reduced by 1.35, 0.59, and 0.37 M, which corresponds to reductions of 59%, 39%, and 28%, respectively. Furthermore, our model differs in accuracy from the most accurate model, MobileNetV3, by only 0.4%, which is perfectly acceptable; we make up for the disadvantage of MobileNet-RepMLP in terms of accuracy by introducing the ECA module and Hardswish activation function. With the introduction of the ECA module and the Hardswish activation function, our model achieves an accuracy of 99.53%. Compared with the ResNet18, MobileNetV2, ShuffleNetV2, and MobileNetV3 models, our model has an accuracy that is improved by 0.42, 0.30, 0.39, and 0.13 percentage points, respectively, and the number of parameters is only 0.91 M. Compared with the optimized ShuffleNet-ECANet 2.0 model, our model has an accuracy improvement of 0.19%, and the parameter difference is only 0.04M, which is not significant. However, in terms of computational complexity, the ShuffleNet-ECANet 2.0 model has achieved better results. The final results show that the accuracy, recall, and precision of our model are 99.53%, 99.52%, and 99.53%, respectively. Compared with the improved MobileNetV2 model, the accuracy is improved by 0.3%, and the numbers of parameters and floating-point operations are reduced by 59% and 18%, respectively. Our model thus performs optimally in terms of accuracy and number of parameters when compared with other lightweight models.

Figure 9 Model comparison.

Figure 10 Loss value change curves for each model.

Table 6 Experimental results of model comparison.

Models	Accuracy/%	Recall/%	Precision/%	Param/MB	FLOPs/M	Madd/M	
ResNet18	99.11	99.10	99.10	11.19	1,820.0	3,642.0	
MobileNetV2	99.23	99.23	99.22	2.26	319.0	625.19	
ShuffleNetV2x1.0	99.14	99.13	99.13	1.28	149.6	295.76	
ShuffleNet-ECANet 2.0	99.34	–	–	0.87	94.25	186.99	
MobileNetV3-Small	99.40	99.39	99.40	1.50	58.8	115.7	
MobileNet-RepMLP (Our)	99.00	99.01	98.99	0.91	266.7	512.4	
MobileNet-RepMLP + ECA + Hardswish (Our)	99.53	99.52	99.53	0.91	268.0	512.4	

Using the PlantVillage test set, we also conducted experiments to ascertain the model inference speed. The main comparison is the time taken by the model to recognize each image. Our model can be made faster by reconstructing the parameters in the RepMLP module. That is, the trained model merges the conv and BN layers in the RepMLP module into the FC layer before deployment. In so doing, the accuracy and the number of parameters of the reconstructed model do not change; its main purpose is to improve the inference speed. We named this reconstructed parameter model Re-MobileNet-RepMLP.

The results in Table 7 show that after reconstructing the parameters, the inference time of our model is 5.87 ms per image, which is 6.8% less than the inference time of a single image before reconstructing the parameters. The inference speed of our reconfigured parameter model is slightly behind ShuffleNetV2 and MobileNetV3, but compared with the original model MobileNetV2, we have improved the inference speed by 0.55 ms and reduced the inference time for a single image by 8.5%. The results show that our model is slightly inferior to the ShuffleNetV2 and MobileNetV3 models in terms of floating-point computations and inference speed. However, it achieves good performance in terms of recognition accuracy and number of parameters, and its comprehensive performance is better than that of MobileNetV2 before improvement.

Table 7 Comparison of model inference speed.

Models	Accuracy/%	Param/MB	FLOPs/M	Speed/ms	
MobileNetV2	99.23	2.26	319.0	6.42	
ShuffleNetV2x1.0	99.14	1.28	149.6	5.54	
MobileNetV3-small	99.40	1.50	58.8	5.33	
MobileNet-RepMLP + ECA + Hardswish	99.53	0.91	266.7	6.30	
Re-MobileNet-RepMLP + ECA + Hardswish	99.53	0.91	268.0	5.87	

Conclusions

Intelligent agriculture is the inevitable trend in the transformation and upgrading of the industry, and the development of advanced methods using science and technology is an important way to achieve high-quality production. In this study, using deep-learning image-recognition techniques, we developed the innovative MobileNet-RepMLP model based on the improved MobileNetV2 model for crop disease prevention and control. Our model provides deployment solutions in the field of crop disease identification with the limited computing resources of mobile and edge devices. The MobileNet-RepMLP model proposed in this article makes the following novel contributions.

(1) Compared with the MobileNetV2 model, the MobileNet-RepMLP model has 59% fewer parameters, requires 18% fewer floating-point operations and 16% fewer multiply–add operations, and has 8.5% faster inference, improving both model performance and computational efficiency.

(2) Experimental analysis using the PlantVillage dataset showed that the MobileNet-RepMLP model achieves 99.0% recognition accuracy. The accuracy can reach 99.53% after adding the ECA module and Hardswish activation function, and the data show that the improved model reaches excellent model benchmarks. Overall, the MobileNet-RepMLP model is a good balance of accuracy, number of parameters, and computation and inference speed, and it represents a significant improvement in performance compared to the original model. The MobileNet-RepMLP model will provide a useful reference for the application of crop disease identification in mobile and edge devices.

This study considered the prevention and control of disease in production crops by refining the algorithmic model for crop disease identification. It explored different models in terms of model simplicity, and the experimental phase used the PlantVillage crop leaf disease dataset for testing. However, It should be noted that the environment and conditions used to acquire the photographs in this dataset were controlled. Although we used data-enhancement operations to simulate the variable imaging conditions of real scenes, this still cannot allow the model to sufficiently adapt to the recognition of real-scene images. The current lightweight models perform well in identifying crop diseases in simple backgrounds. If the model is optimized or improved, it will perform better in this scenario. However, future crop disease identification scenarios will certainly face complex situations. It is worth exploring how to make the model perform well in complex scenarios of crop disease identification. Therefore, in the future, we intend to collect crop leaf disease images using complex backgrounds and explore the performance of the model using these images. We will further enhance the feature-extraction capability of the model to make it perform better when facing real-world scenarios for crop disease image recognition.

Supplemental Information

Supplemental Information 1 Supplemental Table.

Click here for additional data file.

Supplemental Information 2 Improved model-related code.

Click here for additional data file.

We would like to thank our professors for their instruction and our classmates for their help.

Additional Information and Declarations

Competing Interests

Author Contributions

Data Availability

The authors declare that they have no competing interests.

Jianbo Lu conceived and designed the experiments, analyzed the data, performed the computation work, authored or reviewed drafts of the article, and approved the final draft.

Xiaobin Liu conceived and designed the experiments, performed the experiments, analyzed the data, performed the computation work, prepared figures and/or tables, and approved the final draft.

Xiaoya Ma analyzed the data, authored or reviewed drafts of the article, and approved the final draft.

Jin Tong performed the experiments, authored or reviewed drafts of the article, and approved the final draft.

Jungui Peng performed the experiments, authored or reviewed drafts of the article, and approved the final draft.

The following information was supplied regarding data availability:

The data is available at Mendeley: J, ARUN PANDIAN; GOPAL, GEETHARAMANI (2019), “Data for: Identification of Plant Leaf Diseases Using a 9-layer Deep Convolutional Neural Network”, Mendeley Data, V1, DOI 10.17632/tywbtsjrjv.1.

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
