# Peer review of "Improved MobileNetV2 crop disease identification model for intelligent agriculture"

_PeerJ Computer Science, doi:10.7717/peerj-cs.1595_

## Round 0.1 · original submission · Major Revisions

Based on the reviewer's comments, I would like to request the authors to substantially revise the paper based on the reviewers' comments with any new set of additional experiments, if required, and submit the revised paper to PeerJ Computer Science.

·

Basic reporting

Paper required proof reading.

Experimental design

It is mentioned that high accuracy with lower parameters is acheived from the proposed model. How the system balance this tradeoff?
This paper proposed MobileNet-RepMLP model with computational efficiency. Also, RepMLP and ECA modules are also introduced,will they increase the computation? -Justify.

Validity of the findings

when taken from image dataset, size can be equal. But, real time images has varied size. How this model resize?

Additional comments

1. Fig 4 clarity should be improved.
2. Experimental parameters can be tabulated.
3. Abstract length can be reduced.
4. It is mentioned that high accuracy with lower parameters is acheived from the proposed model. How the system balance this tradeoff?
5. This paper proposed MobileNet-RepMLP model with computational efficiency. Also, RepMLP and ECA modules are also introduced,will they increase the computation? -Justify.
6. Paper required proof reading.
7. when taken from image dataset, size can be equal. But, real time images has varied size. How this model resize?

Reviewer 2 ·

Basic reporting

The study presents an interesting work on disease identification using an improved lightweight MobileNetV2 model. The authors have evaluated the model on publically available CIFAR-10 and PlantVillage crop disease datasets. However, there are few areas in the manuscript that require improvement. I suggest the following revisions before publication:

1. The paper is well written with professional English throughout
2. The manuscript is not formatted properly according to the journal format. The citation and reference style should be re-checked.
3. Figures and tables are well structured
4. Results and discussion requires improvement.

Experimental design

1. The paper fits into the scope of the journal as it proposes a new light weight model architecture.
2. Material and methods section is well written and research question is well defined.

Validity of the findings

1. Underlying data is clear and public benchmark dataset is utilized in the experiments.
2. Conclusion needs to be modified.

Additional comments

The study presents an interesting work on disease identification using an improved lightweight MobileNetV2 model. The authors have evaluated the model on publically available CIFAR-10 and PlantVillage crop disease datasets. However, there are few areas in the manuscript that require improvement. I suggest the following comments to be addressed before publication:

1. The manuscript is not well formatted according to the journal format. Authors may re-check the format in the manuscript template given in the journal website.
2. Why the did the authors choose MobileNetV2 model even though other MobileNet models like MobileNetV3 is available and proven to be more accurate?
3. The model was trained with SGD and learning rate 0.1. The authors have done the hyperparameter tuning? Authors may add a table on the hyperparameter tuning and corresponding performance improvement. Also, discuss it in the results section.
4. The authors should include the loss function graph with epoch during the training to check the convergence, indicating that the DL models converge without over fitting or under fitting.
5. Results and discussion should be modified by comparing the model performance with similar light weight models developed by other studies.
6. The references given in the conclusion section should be removed and that can be better discussed in the last section of the results and discussion.

Reviewer 3 ·

Basic reporting

All images need to be upscaled. Especially the images that contain small fonts. Use the proper color contrast selection, making it easier to understand and enjoyable to learn.

In Figure 9, the performance of each model cannot be distinguished properly because they overlap too much. It would be better for the reader if presented with another graphical model or limited to only the best three models displayed.

Experimental design

no comment

Validity of the findings

no comment

Additional comments

Overall, the article is appropriate and suitable for publication.

---

## Round 0.2 · accepted · Accept

Based on the reviewer's responses, the author(s) have addressed all the major and minor comments. Now the paper can be accepted and ready for publication.

·

Basic reporting

paper is revised as per comments

Experimental design

good

Validity of the findings

good

Additional comments

paper is revised as per comments

Reviewer 2 ·

Basic reporting

Clear and unambigous

Experimental design

Original and fits the scope.

Validity of the findings

Well stated conclusions.

Additional comments

Accepted without any additional comments.